# JOINED : Prior Guided Multi-task Learning for Joint Optic Disc/Cup Segmentation and Fovea Detection

**Huaqing He**[†1]                                          12132116@MAIL.SUSTECH.EDU.CN
[1] *Department of Electrical and Electronic Engineering, Southern University of Science and Technology, Shenzhen, China*

**Li Lin**[†1,2]                                                    LINLI@EEE.HKU.HK
[2] *Department of Electrical and Electronic Engineering, The University of Hong Kong, Hong Kong, China*

**Zhiyuan Cai**[1]                                          CAIZHIYUAN2017@GMAIL.COM
**Xiaoying Tang**[*1]                                        TANGXY@SUSTECH.EDU.CN

**Editors:** Under Review for MIDL 2022

## Abstract

Fundus photography has been routinely used to document the presence and severity of various retinal degenerative diseases such as age-related macula degeneration, glaucoma, and diabetic retinopathy, for which the fovea, optic disc (OD), and optic cup (OC) are important anatomical landmarks. Identification of those anatomical landmarks is of great clinical importance. However, the presence of lesions, drusen, and other abnormalities during retinal degeneration severely complicates automatic landmark detection and segmentation. Most existing works treat the identification of each landmark as a single task and typically do not make use of any clinical prior information. In this paper, we present a novel method, named JOINED, for prior guided multi-task learning for joint OD/OC segmentation and fovea detection. An auxiliary branch for distance prediction, in addition to a segmentation branch and a detection branch, is constructed to effectively utilize the distance information from each image pixel to landmarks of interest. Our proposed JOINED pipeline consists of a coarse stage and a fine stage. At the coarse stage, we obtain the OD/OC coarse segmentation and the heatmap localization of fovea through a joint segmentation and detection module. Afterwards, we crop the regions of interest for subsequent fine processing and use predictions obtained at the coarse stage as additional information for better performance and faster convergence. Experimental results reveal that our proposed JOINED outperforms existing state-of-the-art approaches on the publicly-available GAMMA, PALM, and REFUGE datasets of fundus images. Furthermore, JOINED ranked the 5th on the OD/OC segmentation and fovea detection tasks in the GAMMA challenge hosted by the MICCAI2021 workshop OMIA8.

**Keywords:** Multi-task Learning, Optic Disc and Cup Segmentation, Fovea Localization, Coarse to Fine.

---

† Contributed Equally
\* Corresponding Author

## 1. Introduction

In clinical practice, retinal fundus images have been widely used to diagnose various eye diseases such as glaucoma (Almazroa et al., 2015). In retinal images, the optic disc (OD), optic cup (OC), and fovea are key anatomical landmarks providing important biomarkers for the diagnosis of various eye diseases (Cheng et al., 2021). For example, the vertical Cup-to-Disc Ratio (vCDR) is a measure that is commonly employed to identify glaucoma (Vismay et al., 2018). The macula lies in the central part of the retina, and fovea is identified as the center of the macula (Veena et al., 2020). Fovea is the most sensitive area of vision, which is responsible for sharp central vision. Any lesions occurring near fovea may result in vision damages or even blindness. Therefore, accurate OD/OC segmentation and fovea detection are of great significance for disease evaluation and diagnosis (Li et al., 2021; Peng et al., 2021; Lin et al., 2021).

A plentiful of works have been proposed to segment OD and/or OC in fundus images, which can be mainly divided into traditional image processing based methods (Elbalaoui et al., 2018; Sarathi et al., 2016; Park et al., 2006) and recent deep learning based methods (Gao et al., 2020; Manjunath et al., 2020; Vismay et al., 2018). However, according to a recent study, deep learning techniques have dominating superiority on this OD/OC segmentation task (Veena et al., 2020). During the past several years, a variety of deep learning based OD/OC segmentation methods have been proposed. For example, Manjunath et al. designs a residual encoder-decoder network instead of the typically-employed fully convolutional network to boost the OD/OC segmentation performance (Manjunath et al., 2020). In addition to exploring the network structure, Xie et al. also employs a coarse-to-fine strategy to continuously adjust the segmentation region of interest (ROI) to achieve better performance (Xie et al., 2021). Other methods also make use of additional information to assist the OD/OC segmentation task. For example, Vismay et al. feeds coordinate information of OD/OC into a neural network as additional inputs to effectively learn the OD/OC structure (Vismay et al., 2018). Fu et al. makes use of the prior information that OD spatially contains OC to turn the segmentation task into a layered problem (Fu et al., 2018). However, these methods typically miss important prior information from other anatomical landmarks such as the blood vessels and the fovea.

The fovea location is very useful prior knowledge for OD/OC segmentation. Some previous works have already used fovea localization as an auxiliary task to improve the performance of OD/OC segmentation (Huang et al., 2020; Kamble et al., 2020; Meyer et al., 2018). Kamble et al. proposes a two-stage approach combining OD/OC and fovea segmentation (Kamble et al., 2020). Meyer et al. proposes a new strategy for jointly detecting OD and fovea based on distance information (Meyer et al., 2018). Some methods make use of the relative position between OD/OC and fovea to improve the segmentation and localization performance (Huang et al., 2020; Bhatkalkar et al., 2021). A representative work is that Bhatkallar et al. uses heatmap regression to localize OD/OC and fovea, and the performance is competitive with state-of-the-art (SOTA) methods (Bhatkalkar et al., 2021).

Inspired by these aforementioned works, we propose a multi-task learning framework for OD/OC segmentation and fovea detection, named JOINED. The proposed JOINED consists of a coarse stage and a fine stage. At the coarse stage, we design a joint segmentation

and detection module (JSDM) to obtain coarse OD/OC segmentation and fovea location. At the fine stage, we propose two guiding modules for OD/OC segmentation and fovea localization, namely the fine segmentation module (FSM) and the fine localization module (FLM). Different from some coarse-to-fine methods (Kamble et al., 2020; Vismay et al., 2018), we concatenate the output obtained at the coarse stage with the original fundus image as the input of the fine stage, leading to faster convergence and better accuracy.

The main contributions of this work are three-fold: (1) To yield robust outputs at the coarse stage, we propose a multi-task learning model for joint OD/OC segmentation and fovea detection, and design a distance prediction branch to make better use of clinical prior knowledge. (2) At the fine stage, on the basis of the outputs from the coarse stage, we design a multi-branch fovea localization module employing coordinate regression and heatmap detection. To refine the OD/OC segmentation, we use the coarse segmentation output as an additional input, boosting the efficiency and accuracy of the two tasks of interest. (3) We evaluate our proposed JOINED on three publicly-available fundus datasets, and experimental outputs show our approach achieves SOTA performance in both tasks. We rank the 5th in the GAMMA[1,2] (Wu et al., 2022) challenge hosted by the MICCAI2021 workshop OMIA8. We make our code available at https://github.com/HuaqingHe/JOINED.

## 2. Method

### 2.1. Problem setting and model overview

Given an input retinal fundus image $I \in R^{H \times W \times C}$ , where $H$ and $W$ are the height and width of the image, and $C$ is the number of input channels, our goal is to incorporate prior knowledge for better OD/OC segmentation and fovea detection. To this end, we design a novel framework, namely JOINED, that consists of three modules: JSDM, FLM for fovea detection, and FSM for OD/OC segmentation. There are two stages in JOINED, namely a coarse stage and a fine stage. An overview of our proposed JOINED pipeline is shown in Figure 1.

The goal of a segmentation task is to estimate the corresponding segmentation mask $M$. Assume we are given a training set $T = \{I^i, \mathcal{D}^i, M^i, H^i\}_{i=1}^N$, where $\mathcal{D}$ represents distance map from the center of OD and fovea to other positions, $H$ is the heatmap constructed with the coordinates of fovea and the center of OD through a Gaussian kernel matrix $\mathcal{G}(\cdot)$, and $M$ is the ground truth segmentation of $I$. For fovea detection, our goal is to get a coordinate $C = [X, Y]$.

### 2.2. JOINED

We now describe the three network modules of our JOINED framework in detail as below.

**Joint Segmentation and Detection Module**  JSDM contains three decoder branches that share a common encoder, $[\mathcal{D}_P, H_D, P_S] = \mathcal{F}_{JSDM}(I; \theta_{JSDM})$, where $\theta_{JSDM}$ denotes the parameters of $\mathcal{F}_{JSDM}$, $\mathcal{D}_P$ is the predicted distance map, $H_D$ is the output of the detection branch and $P_S$ is a probability map generated by the segmentation branch.

---

1. https://gamma.grand-challenge.org/
2. https://aistudio.baidu.com/aistudio/competition/detail/90/0/submit-result

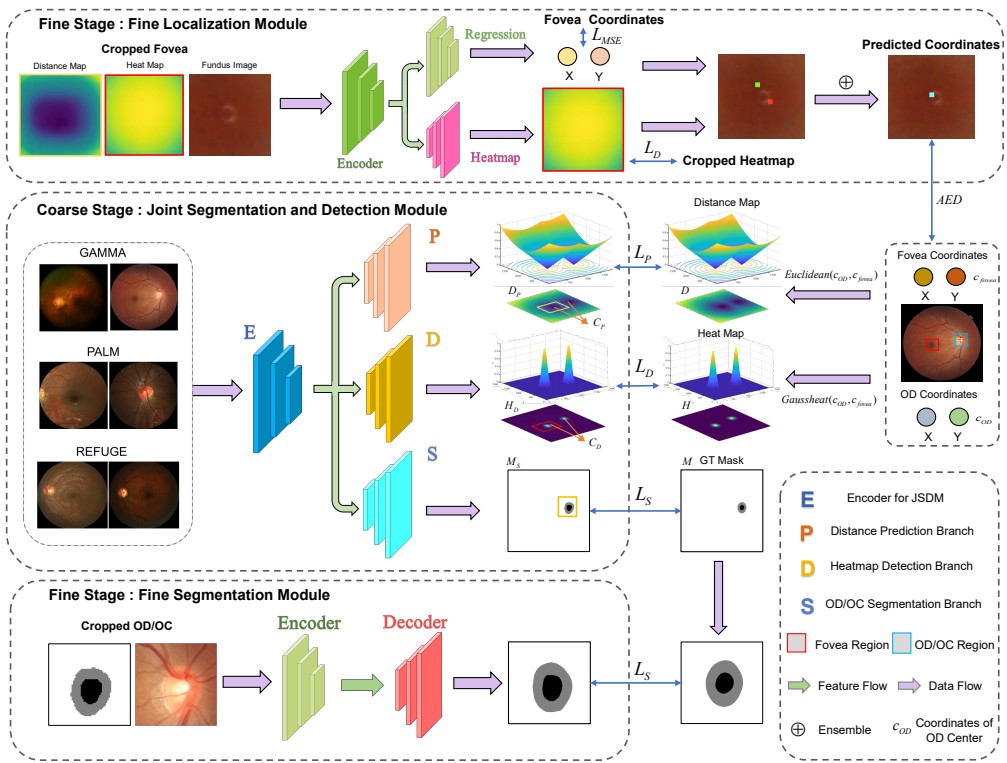

Figure 1: Overview of the proposed JOINED framework.

The distance prediction branch produces a distance matrix $\mathcal{D}_P$ characterizing a global distribution of distances across the entire image. With this auxiliary branch, the coarse detection and segmentation process will be more stable. The distance prediction branch is trained using the Mean Squared Error (MSE) loss,

$$\mathcal{L}_P = MSE(\mathcal{D}, \mathcal{D}_P). \tag{1}$$

In the detection branch, the outputted heatmap $H_D$ has two layers, one of which represents the heat area for OD/OC and the other for fovea (Thewlis et al., 2019). If the localization task does not proceed very well, we will approximate $C$ through a pre-specified relationship between fovea and the center of OD following a previously-published work (Huang et al., 2020). We accumulate each layer according to the coordinate axis and identify indices corresponding to the maximum value as the coordinates of fovea and the center of OD (Li et al., 2020). We compare the coordinates $\boldsymbol{c}_D$ obtained by the detection branch with the coordinates $\boldsymbol{c}_P$ obtained by the prediction branch, as an additional mutually consistent constraint for the detection task. The objective function $\mathcal{L}_D$ of the detection branch is defined in Equation (2), where $M_H$ are the integers of $H$ through thresholding at 0.5; $M_H$ is the mask of OD and macula regions. We employ Dice loss to be the segmentation branch's loss, as defined in Equation (3),

$$\mathcal{L}_D = MSE(H, H_D) + MSE(\boldsymbol{c}_P, \boldsymbol{c}_D) + Dice(M_H, H_D). \tag{2}$$

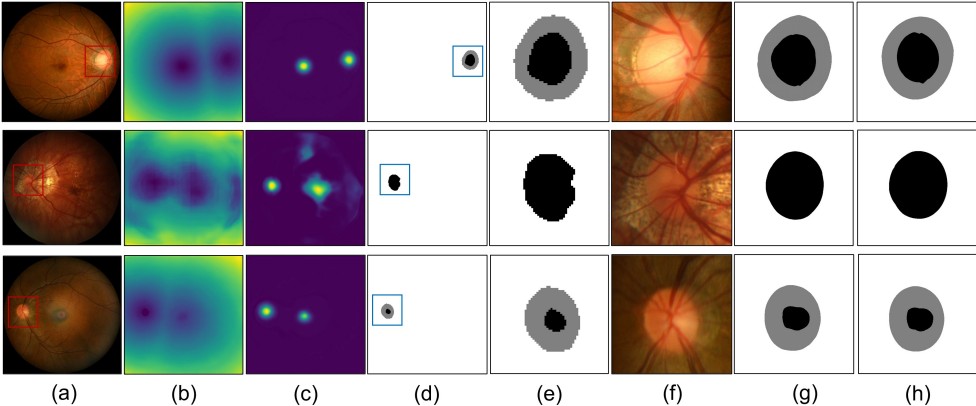

Figure 2: Visualization of the the JSDM outputs obtained on representative images from the three datasets. (a) Retinal fundus images; (b) distance maps obtained from the prediction branch; (c) heatmaps obtained from the detection branch; (d) coarse segmentation results obtained from the segmentation branch; (e) cropped coarse segmentation results; (f) cropped fundus images; (g) fine segmentation results; (h) ground truth segmentation results. From top to bottom are representative cases from the GAMMA, PALM and REFUGE datasets.

$$\mathcal{L}_S = Dice(M, P_S) = 1 - \frac{2\sum MP_S}{\sum (M + P_S) + \varepsilon}. \tag{3}$$

In the training phase, the distance prediction branch is first utilized to extract global semantics of the fundus images. We set a starting flag $\tau_0$ when the branch $P$ almost converges to start the training of the detection branch. Then the heatmap detection branch is utilized to detect the approximate locations of the OD and fovea areas to help more stable OD/OC segmentation. Similarly, we set another starting flag $\tau_1$ when the branch $D$ is close to convergence, to start the training of the segmentation branch. Therefore, the final loss of JSDM is progressively defined as (with coefficients $\lambda_0$, $\lambda_1$ used to balance the three terms)

$$\mathcal{L}_{JSDM} = \begin{cases} \mathcal{L}_P, & \text{epoch} \leq \tau_0 \\ \mathcal{L}_P + \lambda_0 \mathcal{L}_D, & \tau_0 < \text{epoch} \leq \tau_1 \\ \mathcal{L}_P + \lambda_0 \mathcal{L}_D + \lambda_1 \mathcal{L}_S, & \text{epoch} > \tau_1 \end{cases} \tag{4}$$

Detailed network configurations are presented in Appendix A.

**Fine Segmentation Module** FSM utilizes an adapted UNet (Ronneberger et al., 2015) structure with EfficientNet-B4 (Tan and Le, 2020) pretrained on ImageNet (Hagos and Kant, 2019) as the encoder, which produces the final segmentation result $M_{FSM}$. Inspired by previous works (Kamble et al., 2020; Xie et al., 2021), we not only input the OD/OC ROI that is initially segmented and positioned by the JSDM segmentation branch into FSM, but also concatenate the segmentation output $M_S$ obtained in the coarse stage with the original fundus image. Note that $M_S$ is the integer version of $P_S$ through thresholding at 0.5. We feed the concatenated data to FSM for faster convergence and better accuracy. The loss function of FSM is the same as that in the segmentation branch. Figure 2 shows

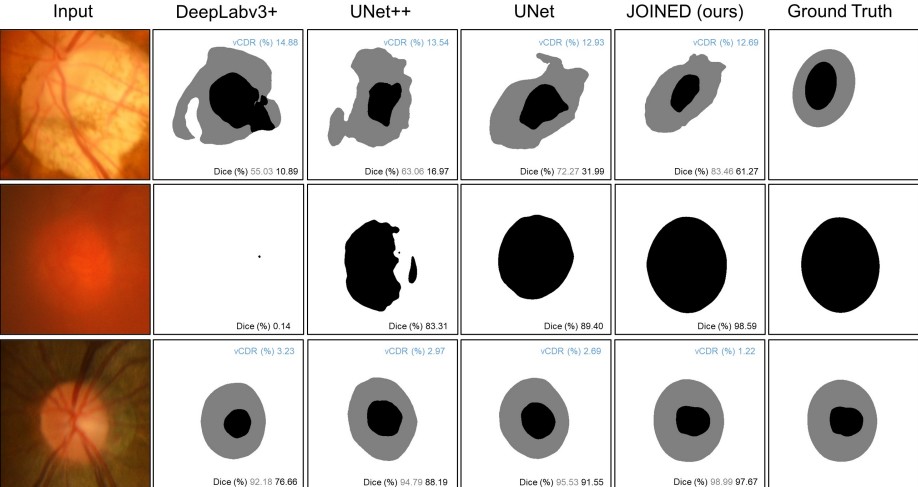

Figure 3: Representative segmentation results from different automated methods and the ground truth. From top to bottom are representative cases from the GAMMA, PALM and REFUGE datasets. For the GAMMA and REFUGE data, the gray area represents OD and the gray number represents the corresponding Dice score of OD; the black area represents OC and the black number represents the corresponding Dice scores of OC; the blue number represents the corresponding vCDR score. For the PALM data, the black area represents OD and the black number represents the corresponding Dice score of OD.

representative visualization results obtained from each branch and the fine-tuned results from FSM.

**Fine Localization Module** A multi-task learning strategy is adopted in FLM. Specifically $[c, H_{FLM}] = \mathcal{F}_{FLM}(I_{crop}, D_{crop}, H_{crop}; \theta_{FLM})$, where $c$ is the predicted coordinates of fovea and $H_{FLM}$ is the predicted heatmap of the cropped fovea area. We concatenate $H_D$ and $\mathcal{D}_P$ from the coarse stage with the original fundus image as the input. We employ FLM to produce predicted fovea coordinates as well as accurate estimates of $H_{FLM}(x, y)$ simultaneously for all pixels and ensembling is adopted to get the final coordinate $\hat{c}$. The loss function is defined as in Equation (5) and $c_{fovea}$ is the ground truth of $\hat{c}$. The finally predicted coordinates of fovea are an ensemble of $c$ and the coordinates obtained from $H_{FLM}$.

$$\begin{aligned} \mathcal{L}_{FLM} &= \mathcal{L}_{regression} + \mathcal{L}_{heatmap} \\ &= MSE(c_{fovea}, \hat{c}) + MSE(H_{crop}, H_{FLM}). \end{aligned} \quad (5)$$

## 3. Experiments and Results

We evaluate our proposed JOINED on three retinal fundus image datasets for OD/OC segmentation and fovea detection: GAMMA, PALM, REFUGE. On each dataset, we compare our method with representative SOTA methods. Due to page limit, we present results on GAMMA in Table 1 and results on PALM and REFUGE in Appendix B.

## 3.1. Implementation details

The proposed JOINED pipeline is implemented with Pytorch, using NVIDIA GeForce RTX 2080Ti GPUs. We use ResNeSt50 (Zhang et al., 2020) as the encoder for JSDM and EfficientNet-B4 (Tan and Le, 2020) for both FSM and FLM. We use the Adam optimizer with a learning rate of $2 \times 10^{-4}$. In our experiments, we set the starting points of joint learning $\tau_1$ as 50, $\tau_2$ as 100. The trade-off coefficients $\lambda_0$, $\lambda_1$, and $\sigma$ are set to be 1, 1, and $H/100$. The estimated total size of our model is 4343.25 MB with 101.9M parameters. The training time is about 48 hours for 300 epochs on GAMMA and 72 hours for 300 epochs on both PALM and REFUGE. The test time is about 0.5 seconds for $2992 \times 2000$ image.

## 3.2. Comparison to SOTA

All methods are assessed with five metrics, i.e., Average Euclidean Distance (AED, pixel) in terms of both OD center and fovea, Dice (%) of both OD and OC, Mean Absolute Error (MAE) in vCDR (%). In Table 1, we compare JOINED against several baseline segmentation models, including UNet (Ronneberger et al., 2015), UNet++ (Zhou et al., 2018) and Deeplabv3+ (Chen et al., 2018) as well as six top-ranking deliveries (other than ours) in the GAMMA challenge. We further replace the encoders of the three baseline methods with ResNet50 and add a branch that outputs coordinates. We provide the implementation details of the three improved baseline methods in Appendix B.1.

Table 1: Performance comparisons between our proposed JOINED and other SOTA methods including three baseline segmentation models (UNet, UNet++, DeepLabv3+) and six top-ranking deliveries in the GAMMA challenge, as evaluated on the GAMMA dataset.

| Method | GAMMA | | | | |
| | Detection | | Segmentation | | |
| | Fovea AED ↓ | OD AED ↓ | OD Dice (%) ↑ | OC Dice (%) ↑ | vCDR (%) ↓ |
|---|---|---|---|---|---|
| Rank #1 | 13.20 | - | 95.77 | 88.06 | 3.803 |
| Rank #2 | 13.08 | - | 95.85 | 87.68 | 3.700 |
| Rank #3 | 15.75 | - | 95.48 | 87.58 | 3.954 |
| Rank #4 | 13.32 | - | 95.6 | 87.98 | 4.129 |
| **Proposed** | 15.15±30.56 | 22.93±28.84 | 95.53±5.60 | 86.89±9.10 | 3.938±2.24 |
| Rank #6 | 15.75 | - | 95.83 | 87.76 | 4.174 |
| Rank #7 | 15.84 | - | 95.15 | 87.22 | 4.012 |
| UNet (ResNet50) | 39.27±84.14 | 35.14±48.52 | 91.28±7.38 | 79.46±9.87 | 12.75±9.47 |
| UNet++ (ResNet50) | 43.15±162.56 | 32.93±178.84 | 89.94±6.58 | 73.24±21.43 | 14.43±10.24 |
| DeepLabv3+ (ResNet101) | 46.67±146.28 | 31.93±171.59 | 88.45±9.38 | 75.46±14.69 | 15.45±11.39 |
| UNet (Ronneberger et al., 2015) | - | - | 87.15±10.87 | 76.54±23.90 | 14.21±11.24 |
| UNet++ (Zhou et al., 2018) | - | - | 85.41±15.27 | 75.09±24.25 | 16.93±15.22 |
| DeepLabv3+ (Chen et al., 2018) | - | - | 86.22±12.25 | 72.80±26.15 | 15.24±10.81 |

Apparently, our proposed JOINED outperforms all the three baseline segmentation models (UNet, UNet++, DeepLabv3+) by very large margins. Compared to the top-ranking deliveries on the GAMMA challenge, JOINED is comparable. The ranking was established based on a combined score of all evaluation metrics. Overall, JOINED ranks the 5th. Since the top four methods in the leaderboard were not published yet, JOINED holds SOTA among all published methods. Furthermore, it is worth pointing out that JOINED ranks the 3rd when assessed by the vCDR metric which is a very critical index for clinical diagnoses of glaucoma (Jonas et al., 2000). Representative visualization results from JOINED

on GAMMA, PALM and REFUGE are shown in Figure 3. Clearly, the segmentation results of both OD and OC produced by JOINED are more precise and more accurate than those produced by other compared methods. Representative segmentation results for low-quality images from GAMMA and PALM (row 1 and row 2) as well as high-quality images from REFUGE (row 3) are presented in that figure. Our proposed JOINED yields the highest Dice and vCDR scores in all cases.

### 3.3. Ablation study

The proposed multi-task learning framework is composed of three branches, including a distance prediction branch $P$, a detection branch $D$ and a segmentation branch $S$. To verify the contribution of each of them, we construct five variants of JOINED and conduct ablation studies on the aforementioned three datasets. The ablation analysis results on the GAMMA dataset are tabulated in Table 2. Model I and model II are respectively the baseline segmentation network and the baseline detection network. Model III consists of both the detection branch and the segmentation branch. It shows that when these two tasks are performed together, the performance of fovea localization gets improved. Although the segmentation accuracy slightly decreases, the stability of the segmentation is enhanced (as evaluated by the standard deviation). Model IV and model V respectively show the benefits of the distance prediction branch exerted to the segmentation task and the detection task. Finally, the best results are obtained when all three branches are included (our proposed JOINED). Although there is slight drop in the segmentation accuracy, the stability and other indicators are improved.

Table 2: Ablation analysis results on the GAMMA dataset.

| Model | Component | | | GAMMA | | | | |
| | Predictor | Detector | Segmentor | Detection | | Segmentation | | |
| | | | | Fovea AED ↓ | OD AED ↓ | OD Dice (%) ↑ | OC Dice (%) ↑ | vCDR (%) ↓ |
|---|---|---|---|---|---|---|---|---|
| I | | | ✓ | - | - | 95.31±6.57 | 86.16±13.68 | 4.94±3.81 |
| II | | ✓ | | 20.75±33.13 | 26.34±35.12 | - | - | - |
| III | | ✓ | ✓ | 16.28±33.21 | 24.00±30.91 | 94.98±6.17 | 85.48±10.51 | 5.33±3.12 |
| IV | ✓ | | ✓ | - | - | **95.80±5.93** | **87.17±12.10** | 4.09±2.53 |
| V | ✓ | ✓ | | 18.22±32.11 | **22.83±29.57** | - | - | - |
| **Proposed** | ✓ | ✓ | ✓ | **15.15±30.56** | 22.93±28.84 | 95.53±5.60 | 86.89±9.10 | **3.938±2.24** |

## 4. Conclusion

In this paper, we proposed and validated JOINED, a novel prior guided multi-task and distance aware joint learning framework for OD/OC segmentation and fovea localization. Specifically, by constructing a heatmap detection branch and a distance prediction branch, we incorporated the distance information from all pixels of the fundus image to the two key landmarks (OD/OC and fovea) in the network. We also designed a strategy to obtain more stable outputs by ensembling outputs from coordinate regression and heatmap detection. Extensive experiments on three publicly accessible retinal fundus datasets show that JOINED significantly outperformed SOTA methods on both segmentation and detection tasks, exhibiting the effectiveness of the distance prior knowledge and the joint learning strategy.

## Acknowledgments

This study was supported by the National Natural Science Foundation of China (62071210), the Shenzhen Basic Research Program (JCYJ20190809120205578), the National Key R&D Program of China (2017YFC0112404), and the High-level University Fund (G02236002).

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

## Appendix A. The Joint Segmentation and Detection Module

We incorporate prior information of explicit and implicit topology into the encoder by creating an auxiliary task to help identify the OD/OC center and the fovea. We use EfficientNet-B4 (Tan and Le, 2020) as the encoder for both FSM and FLM. The Adam optimizer with a learning rate of $2 \times 10^{-4}$ is employed. The encoder consists of five encoder blocks, and the outputs of each encoder block are passed through a maxpooling layer, with a kernel size of 2, before being forwarded to the next encoder block. Moreover, we use long skip connections to connect the feature maps of the first four blocks of the encoder to the corresponding features of each decoder. It not only recovers the spatial information lost during downsampling, but also enables feature's reusability and stabilizes training and convergence. Three small decoders, namely Predictor, Detector and Segmentor, are used in our setting, with the number of feature maps starting at 256 and getting halved after each layer of upsampling. The features obtained from the penultimate layer of the Predictor are connected to those obtained from the corresponding layer of the Detector to better perceive the position information of the macula and OD. Each decoder module comprises nearest upsampling with a scale factor of 2, followed by two layers of 3×3 filters, batch normalization (BN), and ReLU.

### A.1. The distance map prediction branch - Predictor

In this branch, we perform a distance map prediction task. The map $\mathcal{D}$ is generated from the coordinates of OD center $\boldsymbol{c_{OD}}$ and fovea $\boldsymbol{c_{fovea}}$. The coordinate of the OD center $\boldsymbol{c_{OD}}$ is defined as

$$\boldsymbol{c_{OD}} = \begin{bmatrix} x_{od}, y_{od} \end{bmatrix} = \frac{\max \begin{bmatrix} X_{OD}, Y_{OD} \end{bmatrix} + \min \begin{bmatrix} X_{OD}, Y_{OD} \end{bmatrix}}{2}, \tag{6}$$

where $X_{OD}, Y_{OD}$ are the set of coordinates of all OD pixels in the ground truth segmentation mask $M$. The coordinate of fovea $\boldsymbol{c_{fovea}}$ is manually identified. Each value in $\mathcal{D}$ is defined as the shorter distance from the corresponding pixel's location to $\boldsymbol{c_{OD}}$ or $\boldsymbol{c_{fovea}}$ (Meyer et al., 2018). We thus obtain the ground truth distance map $\mathcal{D}$ as

$$\mathcal{D}(x,y) = \min \left\{ \sqrt{(x - x_{od})^2 + (y - y_{od})^2}, \sqrt{(x - x_{fovea})^2 + (y - y_{fovea})^2} \right\}. \tag{7}$$

Afterwards, we normalize $\mathcal{D}$ to yield $\mathcal{D}^N$ to serve as the ground truth distance map.

$$\mathcal{D}^N(x,y) = 1 - \frac{D(x,y)}{\max D(x,y)}. \tag{8}$$

### A.2. The heatmap detection branch - Detector

In the detection branch, we generate two heatmaps $\mathcal{G}(\boldsymbol{c_{OD}})$ and $\mathcal{G}(\boldsymbol{c_{fovea}})$, through two Gaussian kernel matrices, to represent $\boldsymbol{c_{OD}}$ and $\boldsymbol{c_{fovea}}$. After getting $\mathcal{G}(\boldsymbol{c_{OD}})$ and $\mathcal{G}(\boldsymbol{c_{fovea}})$, we normalize them to [0,1] and concatenate them together to form the detection branch's ground truth $H$. Specifically,

$$\mathcal{G}(\boldsymbol{c_k}) = \frac{1}{2\pi\sigma^2} e^{-\frac{||\boldsymbol{x} - \boldsymbol{c_k}||_2^2}{2\sigma^2}}, k \in \{OD\ center, fovea\}, \tag{9}$$

$$H_D = concatenate\big(\mathcal{G}(\boldsymbol{c_{OD}}), \mathcal{G}(\boldsymbol{c_{fovea}})\big), \tag{10}$$

where $\sigma$ is set to be $H/100$. If there exists no OD or fovea due to poor image quality, we set the value of the corresponding $c_{OD}$ or $c_{fovea}$ to zero to ensure the robustness of the detector. To improve the perception of each ROI, features from the penultimate layer of the predictor are concatenated with those from the equivalent layer of the detector. Noticeably, in Equation (2) we employ both the MSE loss and the Dice loss to improve the performance of the regression task during training.

### A.3. The Segmentation Branch - Segmentor

The segmentor outputs a probability map $P_S \in [0, 1]^{H \times W \times 3}$, wherein the three layers respectively represent the probabilities of OC, OD, and the background. We set the threshold in all layers of $P_S$ as 0.5. Since it is a multi-classification problem, some pixels may fall into the situation of being classified as OD and OC at the same time. Under such circumstances, we set the priority of category classification as OC > OD, since OD always spatially contains OC.

## Appendix B. Datasets and More Experimental Results

### B.1. Implementation details of the three baseline methods

For a detection purpose, we obtain feature maps from the encoder of each CNN (e.g., UNet, UNet++, and DeepLabv3+) and then input them to two fully connected layers to output the coordinates of OD and fovea. We use ResNet50 as the encoder for UNet and UNet++, ResNet101 as the encoder for DeepLabv3+, and initialize with Imagenet's pretrained parameters for better performance. Compared with the original UNet and UNet++, we incorporate a BN layer after each convolution layer in the decoder. In UNet, the batch-size is set to 16, the learning rate is 1e-5, and Dice loss is used as the loss function. In UNet++, the batch-size is set to 16, the initial value of the learning rate is 1e-3. We combine Dice loss with the standard binary cross-entropy (BCE) loss as its loss function. In DeepLabv3+, the batch-size is set to 16, the learning rate is 0.01, and the BCE loss is used as the loss function.

### B.2. GAMMA

The GAMMA dataset were provided by the GAMMA challenge organizers in MICCAI2021 workshop OMIA8. This dataset include 200 fundus image data, each of which contains a 2D retinal fundus image and a 3D Optical Coherence Tomography (OCT) image. The GAMMA challenge includes three tasks: glaucoma classification, OD/OC segmentation, and fovea localization. The images were collected at multiple equipments, inducing diverse image resolutions, ranging from 1956×1934 to 2992×2000. We train our model on the 100 training data and evaluate on the 100 testing data. Five-fold cross-validation is used for fair comparisons.

## B.3. PALM

The pathologic myopia (PALM)[3] dataset were provided by the ISBI 2019 Pathologic Myopia Ophthalmology Competition organizers. It contains 800 training fundus images and 400 testing images. The image resolution is either 1444×1444 or 2124×2056. Noticeably, for some images in this dataset, there exists no OD/OC or no fovea or neither of them. Besides, there is no ground truth segmentation for OC, and thus there is no evaluation of OC on this dataset. Comparisons between our proposed JOINED and other SOTA methods on the PALM dataset are presented in Table 3.

Table 3: Detection and segmentation performance comparisons on PALM.

| Method | PALM | | |
| --- | --- | --- | --- |
| | Detection | | Segmentation |
| | Fovea AED ↓ | OD AED ↓ | OD Dice (%) ↑ |
| DeepLabv3+ (Chen et al., 2018) | - | - | 65.53±35.18 |
| UNet++ (Zhou et al., 2018) | - | - | 76.82±22.66 |
| UNet (Ronneberger et al., 2015) | - | - | 80.59±20.99 |
| DeepLabv3+ (ResNet101) | 156.13±243.52 | 156.37±399.47 | 69.75±32.67 |
| UNet++ (ResNet50) | 140.52±271.15 | 114.08±307.40 | 82.64±23.12 |
| UNet (ResNet50) | 108.35±125.35 | 92.58±182.67 | 92.79±7.76 |
| Pixel-Wise Regression (Meyer et al., 2018) | 51.59±75.98 | 53.72±68.28 | - |
| H-DenseUNet (Li et al., 2018) | - | - | 69.59±35.92 |
| Attention UNet (Oktay et al., 2018) | - | - | 87.76±9.51 |
| Segtran (Li et al., 2021) | - | - | 94.34±4.98 |
| **Proposed** | **40.15±33.75** | **38.28±46.25** | **94.53±6.51** |

## B.4. REFUGE

This dataset were provided by REFUGE[4] (ORL, 2020), as part of MICCAI 2019. There are a total of 400 images for training, 400 for validation and 400 for testing. The resolution for the training data is 2124×2056 and that for the validation and testing data is 1634×1634. Comparisons between our proposed JOINED and other SOTA methods on the REFUGE dataset are listed in Table 4.

Table 4: Detection and segmentation performance comparisons on REFUGE.

| Method | REFUGE | | | | |
| --- | --- | --- | --- | --- | --- |
| | Detection | | Segmentation | | |
| | Fovea AED ↓ | OD AED ↓ | OD Dice (%) ↑ | OC Dice (%) ↑ | vCDR (%) ↓ |
| DeepLabv3+ (Chen et al., 2018) | - | - | 85.66±14.27 | 71.71±28.07 | 16.27±12.37 |
| UNet++ (Zhou et al., 2018) | - | - | 86.15±13.81 | 72.79±26.57 | 14.55±11.87 |
| UNet (Ronneberger et al., 2015) | - | - | 90.22±10.14 | 73.46±27.35 | 14.00±10.71 |
| DeepLabv3+ (ResNet101) | 105.10±127.57 | 132.45±141.24 | 90.03±6.91 | 78.05±22.91 | 15.45±11.39 |
| UNet++ (ResNet50) | 98.15±112.56 | 82.57±128.84 | 92.52±6.18 | 83.71±18.23 | 14.43±10.24 |
| UNet (ResNet50) | 79.27±84.57 | 81.54±98.15 | 94.97±5.38 | 83.91±10.15 | 12.75±9.47 |
| Pixel-Wise Regression (Meyer et al., 2018) | 42.18±57.27 | 34.75±52.10 | - | - | - |
| H-DenseUNet (Li et al., 2018) | - | - | 91.02±7.21 | 80.16±19.12 | 15.99±11.26 |
| Attention UNet (Oktay et al., 2018) | - | - | 94.35±7.35 | 82.84±13.27 | 12.58±9.33 |
| Segtran (Li et al., 2021) | - | - | **96.08±4.25** | **87.22±8.11** | 4.129±3.14 |
| **Proposed** | **30.40±36.71** | **29.53±35.19** | 95.35±6.12 | 86.94±8.84 | **3.831±2.05** |

---

3. https://aistudio.baidu.com/aistudio/competition/detail/86/0/introduction
4. https://refuge.grand-challenge.org/Home2020/

## Appendix C. Data Augmentation Details

We identify the smallest rectangle that contains the entire field of view and use the identified rectangle to crop each fundus image. We then resize all cropped images to 256×256 before being inputted to the network. The augmentation strategy we employ in training JOINED is as follows. The color distortion operation adjusts the brightness, contrast, and saturation of the images with a random factor in [-0.1, 0.1]. Horizontal and vertical flipping as well as rotation operations are applied with a probability of 0.5 and Gamma noise is applied with a random factor in [-0.2, 0.2]. For the resizing operation, we randomly sample in [1/1.1, 1.1] and then times the original size. For cropping outputs from the coarse stage, we empirically identify 448×448 to be an optimal size for OD/OC segmentation and 128×128 for fovea localization, to be used at the fine stage.

## Appendix D. Additional Experimental Results

### D.1. Structure and encoder for the fine stage

In our next experiment, UNet and UNet++ are selected as the baseline structure, and ResNeSt50 and EfficientNet-B4 are employed as the encoder to identify the best combination. The results are shown in Table 5. As suggested by the results, UNet is better than UNet++ and EfficientNet-B4 is better then ResNeSt50 for our OD/OC segmentation and fovea detection tasks. So we choose the combination of UNet and EfficientNet-B4 for our fine stage.

### D.2. Input resolution

The resolution of the input image largely affects the performance of both segmentation and localization. For the GAMMA dataset, Table 6 shows that the OD/OC segmentation performance becomes worse when the resolution reduces from 448× 448 to 384×384. A potential reason is that for some images the 384×384 resolution cannot cover OD, which highlights the importance of maintaining the structural integrity of OD/OC.

Table 5: Detection and segmentation performance with different structure and encoder combinations.

| Method | | GAMMA | | | |
| --- | --- | --- | --- | --- | --- |
| | | Detection | Segmentation | | |
| Structure | Encoder | Fovea AED ↓ | OD Dice (%) ↑ | OC Dice (%) ↑ | vCDR (%) ↓ |
| UNet++ | ResNeSt50 | 23.81±38.71 | 93.53±7.45 | 85.15±11.92 | 5.29±4.14 |
| UNet | ResNeSt50 | 18.47±33.71 | 94.80±5.79 | 85.72±11.03 | 4.043±3.21 |
| UNet++ | EfficientNet-B4 | 20.27±34.70 | 94.67±6.05 | 85.46±10.58 | 4.215±3.47 |
| UNet | EfficientNet-B4 | **15.15±30.56** | **95.53±5.60** | **86.89±9.10** | **3.938±2.24** |

Table 6 also demonstrates that the OD/OC segmentation performance becomes worse when the resolution increases from 448×448 to 512×512. This emphasizes that it is better to use a relatively smaller resolution while ensuring the integrity of OD/OC in FSM. Similar results are also observed on the other two datasets.

Table 6: OD/OC segmentation performance with different input resolutions.

| Resolution | | 384×384 | 448×448 | 512×512 |
|---|---|---|---|---|
| GAMMA | OD Dice (%) ↑ | 94.22±6.52 | **95.53±5.60** | 95.19±5.35 |
| | OC Dice (%) ↑ | **87.15±8.47** | 86.89±9.10 | 85.58±9.67 |
| | vCDR (%) ↓ | 4.53±3.53 | **3.938±2.24** | 4.357±3.34 |
| PALM | OD Dice (%) ↑ | 93.87±7.41 | 94.21±6.48 | **94.53±6.51** |
| REFUGE | OD Dice (%) ↑ | 93.84±7.62 | **95.35±6.12** | 94.17±6.78 |
| | OC Dice (%) ↑ | **88.53±7.51** | 86.94±8.84 | 85.71±9.42 |
| | vCDR (%) ↓ | 4.189±3.58 | **3.831±2.05** | 4.407±3.19 |

As illustrated in Table 7, in FLM, when the input resolution becomes smaller, the regression outputs become better but the heatmap outputs become worse. Considering the balance between the two performances, we choose 128×128 as the input resolution for FLM.

Table 7: Fovea localization performance with different input resolutions.

| Resolution | | 64×64 | 128×128 | 256×256 |
|---|---|---|---|---|
| GAMMA | Regression AED ↓ | **15.13±23.52** | 16.52±25.14 | 23.35±35.35 |
| | Heatmap AED ↓ | 25.37±35.47 | 17.08±31.40 | **16.58±29.67** |
| PALM | Regression AED ↓ | **40.83±27.53** | 41.52±31.14 | 50.37±55.35 |
| | Heatmap AED ↓ | 48.37±39.41 | 45.21±42.48 | **42.58±52.67** |
| REFUGE | Regression AED ↓ | **33.13±29.52** | 33.52±35.15 | 38.35±42.81 |
| | Heatmap AED ↓ | 35.00±30.47 | 31.08±42.10 | **30.59±43.28** |

