# OpenReview forum: "JOINED: Prior Guided Multitask Learning for Joint Optic Disc/Cup Segmentation and Fovea Detection"
_MIDL.io/2022/Conference — MIDL 2022_

### Official Review · Reviewer_qhtq · 2022-01-23

**Confidence:** 3
**Preliminary Rating:** 4

**Summary:**

This paper present a novel method for prior guided multi-task learning for joint OD/OC segmentation and fovea detection, which consists of a coarse stage and a fine stage. Experimental results showed that the proposed method outperforms existing state-of-the-art approaches on the publicly-available three datasets of fundus images.

**Strengths:**

This paper proposed a novel method for multi-task learning of for joint OD/OC segmentation and fovea detection, providing rich diagnostic information for fundus screening. Experimental results showed that the proposed method outperforms existing state-of-the-art approaches on the publicly-available three datasets of fundus images.

**Weaknesses:**

The weakness of this paper may be that the model high model complexity, because more submodels are introduced. The implementation details of the experiments should be clearer.  A detailed model complexity analysis should be presented.

**Deanonymize Review:**

no

**Detailed Comments:**

1.The authors compare to several SOTA methods. However, attention U-net [1] and DenseUnet [2] seems to be missing. I recommend that the authors  include the two methods.

[1] Oktay O, Schlemper J, Folgoc L L, et al. Attention u-net: Learning where to look for the pancreas[J]. arXiv preprint arXiv:1804.03999, 2018.
[2] Li X, Chen H, Qi X, et al. H-DenseUNet: hybrid densely connected UNet for liver and tumor segmentation from CT volumes[J]. IEEE transactions on medical imaging, 2018, 37(12): 2663-2674.


2. It can be seen from Figure 1, each task owns a corresponding sub-model, so it is necessary to analyze the complexity of the model, including the number of parameters and inference time.


**Paper Type:**

methodological development

**Questions To Address In The Rebuttal:**

1. First, the authors should add the comparison experiments mentioned above, including  attention U-net [1] and DenseUnet [2], they are often used in medical image segmentation .
2. A detailed model complexity analysis is presented.

**Special Issue:**

no

---

### Official Review · Reviewer_XZ9r · 2022-01-24

**Confidence:** 4
**Preliminary Rating:** 3
**Recommendation:** Poster

**Summary:**

This article proposes a new multitask learning architecture for the segmentation of Optic Disc / Optic Cup (OD/OC) and for the detection of the fovea. This architecture is composed of three modules : a coarse stage branch called the Joint Segmentation and Detection Module (JSDM), a fine stage OD/OC segmentation branch called the Fine Segmentation Module (FSM) and a fine stage fovea detection branch called the Fine Localization Module (FLM).

The JSDM is composed of one encoder and three parallel decoders, each performing a different task : distance prediction (auxiliary task), fovea detection and OD/OC segmentation. The distance prediction decoder is trained first to extract global spatial information, then, after tau_0 epochs, the detection decoder is added to locate roughly the fovea and OD, and finally after tau_1epochs, the segmentation decoder is added to perform a rough segmentation of the OD/OC.

The FSM takes as input a crop of the rough segmentation mask of the JSDM centered on the OD/OC, and output a finer segmentation based on a UNet-like pretrained on IMageNet.

The FLM takes as input the distance map and fovea detection (as heat map) from the JSDM and the original fundus image, each one cropped on the fovea. The FLM provide the fovea coordinates and a finer fovea heatmap, both of which are supervised by a specific loss. From the fovea coordinates and heatmap, the FLM should output more precise fovea coordinates.

The authors validated this work on three public datasets and performed an an ablative study.
They ranked 5th with this approach to the GAMMA MICCAI 2021 challenge.

**Strengths:**

The method is well described and illustrated.
The work is well motivated and contextualized with respect to previous works
The authors performed in-depths experiments : validation on 3 different public datasets, ablative study of the 3 modules and comparison with state-of-the-art methods through the participation to a MICCAI challenge.
The authors also provide their code, which makes the method very well reproducible.


**Weaknesses:**

The article contains many conflicts of notations which make the article more difficult to read. As it is well written, it is still understandable, but the notations should be reworked (see detailed comments).

Among the huge amount of (more or less good quality) published work on detection / segmentation of retinal images with deep learning in the last decade, the work proposed in this article is of good quality, but it is neither very original nor provide ground-breaking results. The authors ranked 5th to the challenge, which is correct, but it is not very clear how many methods competed (it is very difficult to find information on the result of this challenge online).

Even though the proposed architecture is well motivated, and parts are validated though an ablative study, it is still very complex (three modules with several branches each), for a small performance increase.


**Deanonymize Review:**

no

**Detailed Comments:**

Conflict of the notations :
    - H represents both the ground-truth heatmap and the height of the input fundus image
    - C represents both the ground-truth fovea coordinate and the number of input channels
    - The probability map generated by the segmentation branch of the JSDM is first called P_S (page 3), but is then called M_S in the loss (Eq.3) and in the rest of the paper, except in the appendix A.3. M_S seems more coherent with the notation of the other two decoders.
    -  In the appendix, it seems that c_fovea and c_OD are the ground truth coordinated of the fovea and OD respectively. However, c_fovea is defined as the predicted coordinated of the fovea in the FLM Module description. Moreover, in Eq. 5, the MSE is computed based on c_fovea and c which is not defined

The authors should add the following notations (or their corrected version) to the framework illustration of Figure 1 : D_P, H_D, M_S, D, H, M, I, C_P, C_D

Contrary to the challenge results, the comparison results with the three baseline methods are not very convincing as the authors do not provide the implementation details, such as the values of the hyperparameters, and how they optimized them.


**Final Rating After The Rebuttal:**

4: Weak Accept

**Justification Of The Final Rating:**

I thank the authors for answering my questions and recommendations. The authors clarified the notations in their article and provided enough information on the different network architectures. The updated version of the article is improved, however my concerns on the high network complexity with regard to the low result improvement still hold.

**Paper Type:**

both

**Questions To Address In The Rebuttal:**

- What is the architecture of the FLM ?
- JSDM : Please clarify what is M_H, currently defined as "the integers of H".
- Please explain how the input of the FLM and FSM are cropped. Is it based on the coordinates of the fovea provided by the prediction branch (C_p) or the ones of the detection branch (C_D) of the JSDM ? What is the size of the crop ? How is it determined ?
- FLM : Please clarify what type of ensembling strategy is performed to output the final predicted fovea coordinates.
- Please provide more information on the GAMMA challenge results : the website of the challenge / results, the number of participants...

**Special Issue:**

no

---

### Official Review · Reviewer_o8Vz · 2022-01-24

**Confidence:** 4
**Preliminary Rating:** 4
**Recommendation:** Poster

**Summary:**

The paper introduces a new multi-task learning method for Optic Disc and Cup segmentation and Fovea localization. The method consists of two parts: coarse segmentation/detection and refinement. The coarse consists of "joint segmentation and detection module" and focuses on three tasks: disc/cup segmentation, heatmap regression, and distance prediction. The fine part consists of the localization module and segmentation module. The method is well validated on open datasets, the authors intend to opensource the code.

**Strengths:**

- public data/ opensource code. The authors use the public data for the development and validation, they also demonstrate intentions to open source the code.
- solid validation. The method is compared against a set of baselines as well as competing methods of the GAMMA challenge.
- ablation study
- appendix with detailed descriptions of the data/ additional results

**Weaknesses:**

- lack of novelty. The proposed framework combines a number of methods that are not novel on their own. The framework looks like a set of tricks to improve the score.
- the prior knowledge is mentioned in the paper, including the title. It is not clear what role does it play and how it affects the end results. It's said that the distance prediction branch makes use of the prior information. To my understanding, this is a purely multi-task objective and the method has nothing to do with incorporating prior information. "Prior" mentioned in the title to attract attention,
- description of the detection branch (core part of the framework)  is not self-sufficient and requires reading prior publications of the authors. I would suggest expanding this part with detailed explanations.

**Deanonymize Review:**

no

**Final Rating After The Rebuttal:**

4: Weak Accept

**Justification Of The Final Rating:**

I thank the authors for the detailed response clarifying some of the questions. I appreciate the changes introduced to the appendix as requested by the reviewers. From my previous impression, I think this is an interesting work. I'm keeping the score the same since I had little time (1 day) to reassess the revised paper, which I did very briefly.

**Paper Type:**

methodological development

**Questions To Address In The Rebuttal:**

I would like the authors to address the point concerning the use of prior knowledge in the proposed framework. What is the role of prior in the method? Can you manipulate the prior to get a different result?

**Special Issue:**

no

---

### Official Review · Reviewer_Z6VX · 2022-01-26

**Confidence:** 5
**Preliminary Rating:** 4
**Recommendation:** Poster

**Summary:**

The authors proposed a method for jointly optic disc/cup segmentation and fovea detection on color fundus images. The method is based on a two stage approach: a coarse stage based on a multi-task network and a fine stage for refinement of both the detection and the segmentation tasks using the initial image and the outputs of the coarse stage. The method is widely evaluated on different publicly available datasets and a challenge and compared to different state-of-the-art approaches.

**Strengths:**

- The method provides a joint segmentation of different landmarks in the retina. This is very valuable in the clinic, specially in diagnosis of abnormalities where the relative position to these landmarks correlates with severity levels.

- An ablation study is performed to evaluate the different choices of the proposed architecture

- The performance of the method is carried out in a challenge but also in different publicly available dataset. And the code will be available to the community. This allows reproducibility and objective comparison of the results.

**Weaknesses:**

- The heatmap detection branch is utilized to help more stable OD/OC segmentation. The authors claim in Section 3.3. that "Although the
segmentation accuracy slightly decreases, the stability of the segmentation is enhanced (as evaluated by the standard deviation)." However, this is not so clear from the results shown in Table 2 where the difference in std is minimal. It would be good to further support this claim.

- In the results for the different databases it is clear the detection performance is improved with the proposed method. However, this is not so clear for the segmentation performance, where the improvement is marginal. Could the authors elaborate further on that?

**Deanonymize Review:**

no

**Detailed Comments:**

- The authors indicate that the GAMMA challenge is part of MICCAI 2021 challenges but I could not find it as such. Could you please correct or explain if correct?

- UNet, UNet++, DeepLabv3+ are segmentation networks but they have been also evaluation for detection. Please, explain how detection is achieved.

**Final Rating After The Rebuttal:**

4: Weak Accept

**Justification Of The Final Rating:**

I appreciate the authors' effort to address my suggestions and comments regarding segmentation performance and providing more details about the comparison study. I keep the same rating as the marginal improvement does not justify the method complexity. Also, please correct in the paper that GAMMA is not part of the challenges from MICCAI2021.

**Paper Type:**

both

**Questions To Address In The Rebuttal:**

Please, address the previously addressed issues:

- The heatmap detection branch is utilized to help more stable OD/OC segmentation. The authors claim in Section 3.3. that "Although the
segmentation accuracy slightly decreases, the stability of the segmentation is enhanced (as evaluated by the standard deviation)." However, this is not so clear from the results shown in Table 2 where the difference in std is minimal. It would be good to further support this claim.

- In the results for the different databases it is clear the detection performance is improved with the proposed method. However, this is not so clear for the segmentation performance, where the improvement is marginal. Could the authors elaborate further on that?

- The authors indicate that the GAMMA challenge is part of MICCAI 2021 challenges but I could not find it as such. Could you please correct or explain if correct?

- UNet, UNet++, DeepLabv3+ are segmentation networks but they have been also evaluation for detection. Please, explain how detection is achieved.

**Special Issue:**

no

---

### Meta-Review · Area_Chair_om2M · 2022-02-20

**Recommendation:** Accept (Poster)
**Confidence:** 5

**Metareview:**

The authors have clearly clarified the issues raised by the reviewers. Overall, all reviewers are satisfied with the response given by the authors and are glad to see that the quality of the paper has been improved substantially.

---

### Decision · Program_Chairs · 2022-02-28

Accept